# AutoParLLM: GNN-Guided Automatic Code Parallelization using Large Language Models

## Abstract

Parallelizing sequentially written programs is a challenging task. Even experienced developers need to spend considerable time finding parallelism opportunities and then actually writing parallel versions of sequentially written programs. To address this issue, we present AutoParLLM, a framework for automatically discovering parallelism and generating the parallel version of the sequentially written program. Our framework consists of two major components: i) a heterogeneous Graph Neural Network (GNN) based parallelism discovery and parallel pattern detection module, and ii) an LLM-based code generator to generate the parallel counterpart of the sequential programs. We use the GNN to learn the flow-aware characteristics of the programs to identify parallel regions in sequential programs and then construct an enhanced prompt using the GNN's results for the LLM-based generator to finally produce the parallel counterparts of the sequential programs. We evaluate AutoParLLM on 11 applications of 2 well-known benchmark suites: NAS Parallel Benchmark and Rodinia Benchmark. Our results show that AutoParLLM is indeed effective in improving the state-of-the-art LLM-based models for the task of parallel code generation in terms of multiple code generation metrics. AutoParLLM also improves the average runtime of the parallel code generated by the state-of-the-art LLMs by as high as 3.4% and 2.9% for the NAS Parallel Benchmark and Rodinia Benchmark respectively. Additionally, to overcome the issue that well-known metrics for translation evaluation have not been optimized to evaluate the quality of the generated parallel code, we propose OMPScore for evaluating the quality of the generated code. We show that OMPScore exhibits a better correlation with human judgment than existing metrics, measured by up to 75% improvement of Spearman correlation.

## 1 Introduction

The rise in the number of on-chip cores has led to more frequent development of parallel code. Nevertheless, to unleash the capabilities of multi-core systems, the need for developing parallel programs will continue to grow. However, developing parallel programs is not a trivial task. The communication among cores, effective data sharing among the threads, synchronization, and many other factors need to be considered while crafting parallel programs, which makes the process of developing parallel programs far more complex than serial ones.

HPC communities have published different tools and programming models to ease the process of moving from serial to parallel code. One of the well-established parallel programming models is OpenMP, which is a directive-based programming model that allows users to parallelize sequential code with minimal changes. Most modern compilers recognize and support parallelization through OpenMP constructs. However, even with OpenMP, developers must carefully decide which clauses or directives they need to use. Inappropriate usage of clauses can cause concurrency bugs such as data race or decrease performance. Therefore, tools such as DiscoPoP (Li et al., 2015) were developed that can automatically discover parallelism opportunities and insert OpenMP constructs. However, due to the conservativeness of these tools, they sometimes miss potential parallelization opportunities.

Nowadays, with the advancements in deep learning, various deep learning approaches have been proposed to identify parallelism and advise OpenMP constructs (Chen et al., 2023a; Harel et al., 2023; Shen et al., 2023; Kadosh et al., 2023). In this paper, we propose AUTOPARLLM. An approach that leverages Large Language Models (LLMs) and Graph Neural Networks (GNNs) to produce parallel code. AUTOPARLLM has two main modules: parallelism discovery and parallel code generation. AUTOPARLLM uses Graph Neural Network (GNN) to model flow-aware information, i.e., data-, control- flow, and call graphs. Once a parallelism opportunity is identified, a GNN-guided prompt is engineered using GNN's results to generate parallel code with the help of an LLM. Due to the nature of OpenMP constructs, sometimes the order is important in the predictions, and sometimes not. To better measure the quality of generated OpenMP constructs we provide a new metric in this paper called OMPScore. Our experimental results show that the enhanced designed prompt helps to unleash the potential power of LLMs in generating parallel code out of serial code.

In summary, our paper provides the following key contributions:

- We propose a novel approach, called AUTOPARLLM, leveraging GNNs to guide the automatic parallelization of code using large language models.

- To the best of our knowledge, AUTOPARLLM is the first automatic parallelization tool built on top of GNNs and LLMs, having the benefits of both worlds.

- We evaluate the proposed approach on well-established benchmarks such as NAS Parallel benchmark and Rodinia benchmark.

- We propose a new evaluation measure called OMPScore to assess the quality of generated OpenMP code.

The format of the paper is as follows: In the next section, we will discuss the related works. In section 3, motivation examples will be discussed. Followed by section 4, where our approach is explained in detail along with the OMPScore. Section 5 presents the experimental results. Section 6 discusses ablation study and finally, section 7 concludes the paper.

## 2 RELATED WORKS

**Traditional parallelism assistant tools:**  Some traditional tools are based on static and dynamic analysis for automatic parallelization of sequentially written programs. PLuTo (Bondhugula et al., 2008) analyzes code statically and can optimize programs for parallel execution. It is a polyhedral source code optimizer based on OpenMP (Meadows, 2007). Rose (Quinlan & Liao, 2011) is another static source-to-source compiler infrastructure that also supports automatic parallelization using OpenMP. Both of these tools do not require or use code runtime information while generating parallel counterparts of sequential programs. DiscoPoP (Li et al., 2016) is a dynamic analysis-based parallelism assistant tool. It uses dynamic control-flow analysis and data-dependence profiling for identifying parallel regions in source programs. However, these traditional analysis-based tools have some drawbacks, as pointed out by Chen et al. (2023a), and they miss a lot of parallelism opportunities due to being overly conservative.

**Data-driven approaches:**  With the significant progress in the field of machine learning and deep learning, many have proposed automatic data-driven approaches to identify parallelism opportunities and suggest appropriate constructs. Chen et al. (2023a) has proposed an approach based on graph neural networks and augmented Abstract Syntax Trees (ASTs) to identify parallel loops. Their results show that their GNN-based approach outperforms PragFormer (Harel et al.), which uses Transformers (Vaswani et al., 2017) to discover parallelism opportunities in code. Shen et al. (2021) uses contextual flow graphs with graph convolution neural networks (Kipf & Welling, 2016) to detect parallelism. (Shen et al., 2023) uses a combination of control flow, data flow, and abstract syntax tree to predict parallelism. Also, there are attempts to use IR-based representation for automatic parallelization of ML models (Schaarschmidt et al., 2021). Even though there has been some progress in adapting machine learning techniques to predict parallelism opportunities, little effort has been applied to connect parallelism detection and code generation. In this work, we address this gap. We connect GNNs and LLMs to not only discover parallelism but also generate parallel code. Connecting GNNs to LLMs has been investigated recently (Ghosh et al., 2022; Chen et al., 2023b). However, to the best of our knowledge, we are the first to leverage the result of GNN to guide LLM to generate parallel code out of serial code.

**Metrics for Translation Evaluation:** The BLEU score (Papineni et al., 2002) is a classical metric for evaluating textual similarity in machine translation. It assesses the overlap between sequences of consecutive $n$ words, called n-grams. Meteor (Banerjee & Lavie, 2005) was introduced to address some of the limitations of the BLEU score, such as its tendency to underestimate high-order n-grams. ROUGE (Lin, 2004) encompasses a set of metrics that evaluate various aspects of textual similarity. To evaluate code generation, recent works have introduced metrics like CodeBLEU (Ren et al., 2020) and CodeBERTScore (Zhou et al., 2023). However, none of these metrics have been specifically designed for evaluating the quality of OpenMP constructs in terms of textual similarity.

## 3 MOTIVATION

The remarkable growth of Large Language Models (LLMs) has led us to use them in a variety of fields. Although LLMs are powerful in Question Answering tasks, sometimes they struggle to generate quality responses for tasks requiring domain-specific knowledge. For example, for the task of parallelism discovery, if LLMs are invoked with basic prompts without any background information, very often they can not generate satisfactory results. For example, the code shown in Listing 1 is a parallel loop with a reduction pattern, and it has two reduction variables, `R23` and `T23`. The correct way to parallelize the loop is to add a `reduction` clause with

Listing 1: Reduction loop taken from IS application of NAS Parallel Benchmark

```
for (i = 1; i <= 23; i += 1)
{
    R23 = 0.50 * R23;
    T23 = 2.0 * T23;
}
```

those two variables. Both GPT-3.5 and GPT-4 recognize the loop as parallel, but they fail to detect the `reduction` clause with proper variables. CodeLlama-34B identifies it wrongly as a non-parallel loop. CodeGen-16B detects it as a parallel loop but wrongly identifies the `reduction` variables as `private` variables. The loop in Listing 2 is non-parallel due to having inter-iteration dependencies. However, when the LLMs: GPT-3.5, GPT-4, and CodeGen-16B were prompted about whether this loop could be parallelized using OpenMP or not, all three LLMs wrongly parallelized the loop by adding the `#pragma omp parallel for` clause except for CodeLlama-34B. The loop in Listing 3 is a parallel loop that is recommended to be parallelized using the `private` clause, as the variable `i` needs to be private to each thread. Although loop counters are considered private by default, usually common practice is to add `private` clause explicitly to the loop. However, all four LLMs identify the parallel loop but do not add the `private` clause for variable `i`, and when we checked the ground truth value for this loop, we found that the developers added the `private` clause to variable `i`. All these examples show that in spite of being extremely powerful, LLMs can fail to generate proper parallel codes even for quite simple cases. However, the performance of the LLMs can be improved if the LLMs are given some background information regarding the loop prior to generating parallel codes. That is what motivated us to develop AUTOPARLLM, which improves the performance of LLMs by providing them GNN-based guidance.

Listing 2: Non-parallel loop taken from MG Application of NAS Parallel Benchmark

```
    for (i1 = d1; i1 <= mm1 - 1; i1 += 1)
{

    u[2*i3-t3-1][2*i2-t2-1][2*i1-d1-1] =
    u[2*i3-t3-1][2*i2-t2-1][2*i1-d1-1] +
    0.25 * (z[i3][i2][i1-1] +
    z[i3][i2-1][i1-1] + z[i3-1][i2][i1-1] +
    z[i3-1][i2-1][i1-1]);

}
```

Listing 3: Private loop taken from Heartwall Application of Rodinia-3.1 Benchmark

```
    for (i = public.endoPoints;
    i <= public.allPoints -1; i += 1) {
        private[i].point_no = i - public.endoPoints;
        private[i].in_pointer =
        private[i].point_no * public.in_mod_elem;
        private[i].d_Row = public.d_epiRow;
        private[i].d_Col = public.d_epiCol;
        private[i].d_tRowLoc = public.d_tEpiRowLoc;
        private[i].d_tColLoc = public.d_tEpiColLoc;
        private[i].d_T = public.d_epiT;
}
```

## 4 APPROACH

In this section, we present AUTOPARLLM. An approach that leverages Graph Neural Network to learn the flow-aware characteristics of the programs, such as control flow, data flow, and call flow, to guide LLMs to better generate parallel code by constructing a GNN-guided OMP prompt. Figure 1 shows the overall workflow of AUTOPARLLM. Figure 1(a) shows the training process where we train a GNN model to predict parallelism opportunity as well as the parallel pattern. Figure 1(b) shows that at inference time, GNN is used to create a GNN-guided OMP prompt for the LLM to generate parallel code.

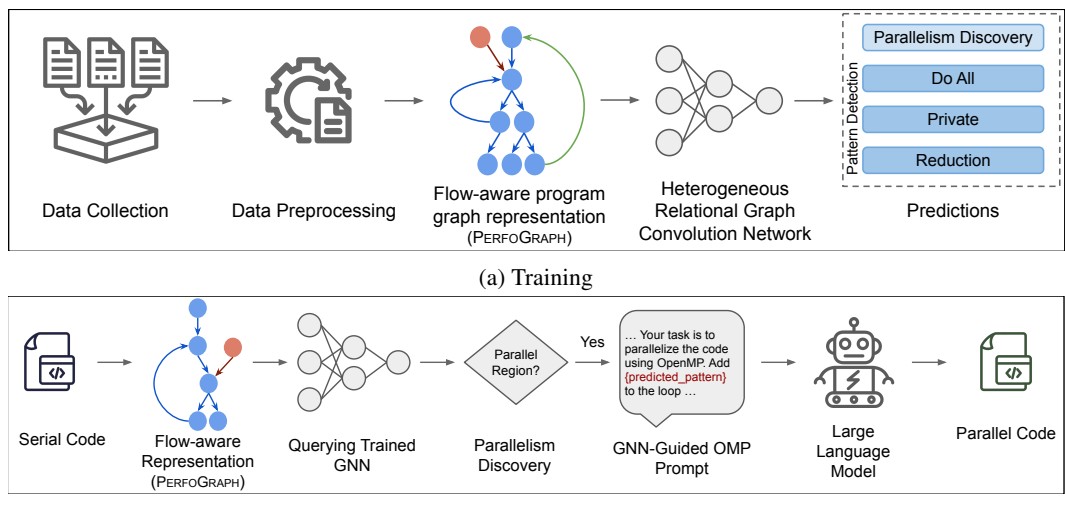

(a) Training

(b) Inference

Figure 1: Overview of the AutoParLLM workflow.

## 4.1 TRAINING

The first step in our approach is training a Graph Neural Network to learn the features of the input programs.

**Data Collection and Preprocessing:** First, we collect data to train our neural network to detect parallelism and patterns. We want our neural network model to be able to realize if a region of a code (such as a loop) is parallelizable or not (for example, due to loop iteration dependency). We use the OMP_Serial dataset (Chen et al., 2023a) for this purpose. Also, some pre-processing is applied to transform the dataset to graph representation of programs so that our GNN-based models can learn efficiently from the representation. The Experimental Results section provides more details regarding the dataset and preprocessing.

**Program Representation:** While different program representations can be used to train neural networks, we use PERFOGRAPH (TehraniJamsaz et al., 2023) program representation in this work. PERFOGRAPH has the advantages of ProGraML (Cummins et al., 2020). Moreover, it can also represent multi-dimensional arrays and vectors in the programs. Additionally, it is numerically aware, meaning it can encode numbers. Experiments that have been conducted on PERFOGRAPH, show that this representation is effective for the task of parallelism discovery and pattern detection (TehraniJamsaz et al., 2023). Also, there is evidence that IR-based representation achieved better results on other downstream tasks like unsupervised code translation as well (Szafraniec et al., 2022).

**Graph Neural Network Training:** In our experiments, we realized it is challenging for Transformer models to easily detect parallel regions or patterns. In the motivation section, we saw that LLMs sometimes can not detect loop dependencies and generate the parallel version of the code where they are not supposed to. Using the PERFOGRAPH representation, we train a Graph Neural Network (GNN) to specifically learn the flow-aware features of the programs. Then, we use our GNN model to predict the parallel regions and patterns. Since there are three types of flow in our program representation, we adapt Heterogeneous Relational Graph Convolution to model and learn each flow in the graph representation.

## 4.2 INFERENCE

In this part, we explain how we utilize the results of our GNN model to guide the LLM model to generate appropriate parallel code.

**Prompt Engineering:** In the first step, before even constructing the prompt for the LLM, we first use GNN model to identify if there is a parallelism opportunity in the given code. If there is no parallelism opportunity, the parallel version of the given code will not be generated. This is an important step, as later in the result section, we will see that some LLMs have difficulties identifying the cases where they should not generate parallel code. Figure 3 shows the GNN-guided OMP prompt which is used to generate parallel OpenMP code. As said, the prompt would be used only if our GNN model predicts a parallelization opportunity. Thereafter, the corresponding patterns will

be predicted by the GNN as well. The supported patterns at the moment are: `do-all`, `private`, `reduction`, and `reduction and private` together. Therefore, the `clause` placeholder in the prompt will be replaced by the name of the predicted pattern.

```
Act as a C++ OpenMP Parallelization Tool.  You will be given a
code.  Your task is to parallelize the code using OpenMP. You
can add any OpenMP clauses if necessary.  If the code is not
parallalizable, output the original given code.\n Code:  {code}\n
Output code:
```

Figure 2: Basic OMP Prompt to generate OpenMP Code

Moreover, we also designed a basic OMP prompt by removing the GNN part of our approach altogether. Figure 2 shows the basic OMP prompt template. It is worth noting that since GNN is not utilized for the basic prompt, whether the code is parallelizable or what kind of pattern exists in the code is delegated to the LLM model itself.

```
Act as a C++ OpenMP Parallelization Tool.  You will be given
a code.  Your task is to parallelize the code using OpenMP and
only output the code.  Add {clause} to the loop.\n Code:  {code}\n
Output Code:
```

Figure 3: GNN-guided OMP Prompt

The choice of LLM depends on the user's preference. Later, in the experimental result section, we will see the results of four different LLMs. Lastly, some LLMs follow a specific prompt template, so our OMP or GNN-guided OMP prompts will be updated to follow the LLM-specific prompt template. For example, Figure 4 shows the GNN-guided OMP prompt template that is used for CodeLlama (Rozière et al., 2023).

```
[INST] <<SYS>> Act as a C++ OpenMP Parallelization Tool.
You will be given a code.  Your task is to parallelize the code
using OpenMP and only output the code.  Add {clause} to the loop.
<</SYS>>\n\n Code:  {code}\n Output Code:  [/INST]
```

Figure 4: GNN-guided OMP prompt for CodeLlama

### 4.3 OMPSCORE

Some characteristics of OpenMP directives and clauses challenge the evaluation using existing textual similarity metrics. To illustrate, consider Figure 5, where we have a candidate and a reference directive, each composed of multiple clauses. One characteristic pertains to the order of variables or operands within certain clauses, where variable rearrangements may not alter semantic meaning (e.g., `private(k,j,i)` is equivalent to `private(j,i,k)`). Another characteristic involves specific clause types where the order of elements significantly affects directive performance. For example, in Figure 5, the candidate directive's `reduction(z:+)` clause is considered a mismatch to the reference directive. In essence, OpenMP directives encompass both order-sensitive and order-insensitive clauses, making a uniform treatment of all clauses as either order-sensitive or order-insensitive inadequate for accurate scoring.

We introduce OMPScore, a metric that enhances Rouge-L score evaluation for OpenMP directives through a combination of regular expressions and program analysis. OMPScore comprises four key modules designed to preprocess input, both candidate and reference directives, to provide improved arguments for Rouge-L score. In the initial stage, the Masking module detects potential clauses or directives for updating. It does so by identifying clauses within OpenMP directives using regular expressions initiated by OpenMP keywords (e.g., `private`, `shared`, or `reduction`) followed by open/close parentheses. Subsequently, the second module categorizes all identified masked spans based on the first word within each span, thereby determining the clause type (e.g., `private`, `shared`, or `reduction`). In the third module, we update the clauses considering two factors related to element order within OpenMP clauses. For instance, the `private` clause in the directive undergoes a sorting action while the reduction clause remains unchanged. To determine whether specific clause types are order-sensitive or order-insensitive, we refer to the official OpenMP doc-

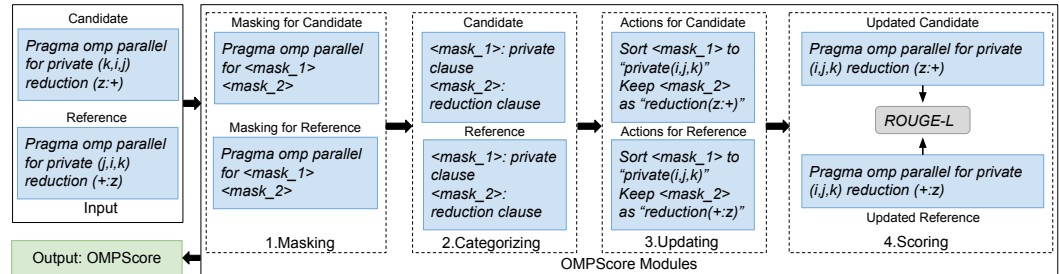

Figure 5: Overview of OMPScore.

umentation and articles[1]. For order-insensitive clause types, like the `private` clause, we alphabetically sort the elements within their respective element lists. Finally, in the fourth module, the updated candidate and reference directives serve as input for the Rouge-L scoring function, yielding the OMPScore, which quantifies the similarity between the candidate and reference directives.

## 5 EXPERIMENTAL RESULTS

We evaluate the effectiveness of AUTOPARLLM on several applications. In this section, we describe the details of those experiments. Also, we describe the components of AUTOPARLLM in detail.

### 5.1 EXPERIMENTAL SETUP

We use the OMP_Serial dataset (Chen et al., 2023a) to pre-train the parallelism discovery and pattern detection module of AUTOPARLLM. The first step is to train our parallelism detection model to identify whether a region, such as a loop, can be executed in a parallel manner. The OMP_Serial dataset contains around 6k compilable C source files that are crawled from Github and well-known benchmarks like PolyBench (Pouchet & Yuki, 2017), Starbench (Andersch et al., 2013), BOTS (Duran et al., 2009), and the NAS Parallel Benchmark (Jin et al., 1999). However, since we use NAS Parallel Benchmark for evaluating the generated parallel codes, we carefully exclude all samples of NAS Parallel Benchmark from the dataset for our pre-training phase so that our model does not "see" those samples beforehand. After excluding those files, LLVM Intermediate Representation (LLVM IR) of source C files is generated. To augment the dataset and increase the size of training data, we compile programs using different LLVM optimization flags following the approach of TehraniJamsaz et al. (2022). Ultimately, we have around 10k IR files (6041 parallel, 4194 non-parallel). The OMP_Serial dataset also contains 200 "private" and 200 "reduction" loops. However, after removing samples taken directly from NAS benchmark and extracted templates, around "158" private and "137" reduction samples are left. Finally, we apply LLVM optimization flags similarly as mentioned above and generate around 4k IR files (2160 private, 2100 reduction). The "private" clause detection model determines the need for a "private" clause. Similarly, the "reduction" clause detection model is used to identify whether we need a "reduction" clause in the OpenMP directive or not. For training "private" clause detection model, two classes are created: private (2160 files) and non-private (contains 2000 files, 50% of those are taken randomly from "reduction" and 50% of those are randomly taken from "non-parallel"). Similarly, for training "reduction" clause detection model, two classes are created: reduction (2100 files) and non-reduction (contains 2000 files, 50% of those are taken randomly from "private" and 50% of those are randomly taken from "non-parallel"). These two models make up the parallel pattern detection module of AUTOPARLLM.

We use the DGL-based (Wang, 2019) implementation of RGCN (Schlichtkrull et al., 2018) with 6 GraphConv layers for all three GNN models. Each source program is a heterogeneous graph represented by PERFOGRAPH (TehraniJamsaz et al., 2023), so the HeteroGraphConv module in each layer is used with the 'sum' aggregation function. We experimented with different hidden layer sizes and learning rates and finally used 64 as the hidden layer size and set the learning rate to 0.01. Each node of the heterogeneous PERFOGRAPH is embedded to a 120-dimensional vector. So, the input layer size is set to 120. The output layer size is set to the number of classes, which is 2, as all three of the GNN models do binary classification. For graph-level prediction, the 'mean' aggregation function combines the results of different node types, and finally linear classifier is used in the last layer of our RGCN model. The linear classifier produce a probability score for each class.

---

[1]https://www.openmp.org/resources/tutorials-articles/

The model is trained for 120 epochs, and the checkpoint with the highest validation accuracy is saved for later inference.

For inference, the three pre-trained models are applied sequentially. First, the input code is passed to the parallelism detection model. If it classifies a loop as parallel, then it is passed to the "private" clause detection model. If the second model classifies it as a "private" loop, then the "private" clause is added to the OMP prompt. Finally, the loop is passed to the "reduction" clause detection model, and similarly, if it classifies the loop as a "reduction" loop, the "reduction" clause is also added to the OMP prompt.

After creating the GNN-guided OMP prompts, the LLMs are invoked to generate the parallel counterpart of the sequential programs. We use four LLMs to demonstrate the performance of AUTOPAR-LLM; note that for the LLMs, the *temperature* parameter is set to zero to make the models deterministic in predicting the OpenMP constructs. We also compare AUTOPARLLM with traditional tools like autoPar and DiscoPoP. However, due to space limitations, we describe those comparisons in the Appendix. We evaluate the performance of AUTOPARLLM on 11 applications of two benchmarks: NAS Parallel Benchmark and Rodinia Benchmark(Che et al., 2009). These applications are developed targeting HPC platforms and heterogeneous computing. Also, both of the benchmarks have OpenMP annotated loops and their sequential version from experienced developers.

## 5.2 EVALUATING CODE GENERATION ON NAS PARALLEL BENCHMARK

First, we evaluate AUTOPARLLM on NAS Parallel Benchmark. We extract loops containing OpenMP pragmas from the eight applications. For extracting loops, we first annotate the loops using the Rose outlining tool (Quinlan & Liao, 2011). Then, we compile and generate the IR for the outlined code. Finally, the `llvm-extract` command is used to extract the loop-specific IR from the full IR. We extracted a total of 454 loops (private: 264, reduction: 17, non-parallel 173). We use 80% of the loops to fine-tune our pretrained GNN models. We train the GNN models in a similar manner described above for 120 epochs. Then we use 20% of loops (90 loops) for evaluating AUTOPARLLM. Of those 90 loops, 58 are parallel, with 56 loops having "private" clause and 2 loops having "reduction" clause. The rest 32 loops are non-parallel. AUTOPARLLM achieves 94.44% accuracy in parallelism discovery by correctly predicting 55 out of 58 parallel loops and 30 out of 32 non-parallel loops. Also, AUTOPARLLM correctly detects 52 out of 56 loops with "private" clause, and it correctly detected all two loops with "reduction" clause.

In Table 1, we compare the performance of the codes generated by using the basic OMP prompt and GNN-guided OMP prompt (denoted as AUTOPARLLM-LLM-name in all tables). We use different score metrics as well as OMPScore for the comparison, and it can be observed that our AUTOPARLLM approach improves all LLMs in terms of these scores.

Table 1: Results on NAS Parallel Benchmark Suite (higher indicates better)

| Model | BLEU | CodeBLEU | Rogue-L | METEOR | CodeBERTScore | ParaBLEU | OMPScore |
|---|---|---|---|---|---|---|---|
| CodeGen-16B | 15.82 | 40.99 | 43.84 | 27.87 | 72.8 | 17.43 | 48.97 |
| AUTOPARLLM-CodeGen-16B | **32.69** | **49.51** | **66.60** | **42.28** | **83.8** | **26.29** | **65.3** |
| GPT-3.5 | 27.11 | 52.57 | 42.83 | 35.16 | 72.3 | 27.44 | 41 |
| AUTOPARLLM-GPT-3.5 | **38.83** | **65.24** | **91.70** | **55.09** | **96.4** | **48.28** | **95.15** |
| GPT-4 | 26.15 | 54.81 | 43.34 | 36.96 | 73.8 | 27.49 | 46.4 |
| AUTOPARLLM-GPT-4 | **40.12** | **68.24** | **90.40** | **55.68** | **95.2** | **48.48** | **95.15** |
| CodeLlama-34B | 26.44 | 53.48 | 45.15 | 36.40 | 74.0 | 26.48 | 45.44 |
| AUTOPARLLM-CodeLlama-34B | **36.14** | **64.81** | **90.59** | **54.86** | **96.0** | **47.73** | **94.46** |

Table 2: Application-wise execution results (in seconds) on NAS Parallel Benchmark Suite. Execution times are average of 5 runs.

| Model | BT | Speedup(%) | IS | Speedup(%) | CG | Speedup(%) | FT | Speedup(%) | LU | Speedup(%) | MG | Speedup(%) | SP | Speedup(%) | Avg. Speedup(%) |
|---|---|---|---|---|---|---|---|---|---|---|---|---|---|---|---|
| GPT-3.5 | 73.560 | | 1.138 | | **0.822** | | 2.270 | | 37.665 | | **1.080** | | 34.163 | | |
| AUTOPARLLM-GPT-3.5 | **71.946** | 2.24% | **1.092** | 4.21% | 0.828 | -0.72% | **2.260** | 0.44% | **37.398** | 0.71% | 1.090 | -0.92% | **30.648** | 11.47% | **2.5%** |
| GPT-4 | 73.424 | | 1.136 | | 0.882 | | 2.250 | | 37.445 | | 1.090 | | 33.648 | | |
| AUTOPARLLM-GPT-4 | **71.892** | 2.13% | **1.087** | 4.51% | **0.880** | 0.23% | **2.244** | 0.27% | **37.052** | 1.06% | **1.080** | 0.93% | **29.322** | 14.75% | **3.4%** |
| CodeLlama-34B | 72.620 | | 1.230 | | 0.872 | | **2.257** | | 37.676 | | 1.090 | | 33.336 | | |
| AUTOPARLLM-CodeLlama-34B | **71.730** | 1.24% | **1.144** | 7.52% | 0.880 | -0.91% | 2.258 | -0.04% | **35.167** | 7.13% | **1.080** | 0.93% | **31.345** | 6.35% | **3.2%** |
| CodeGen-16B | 101.036 | | 1.261 | | 0.914 | | 2.280 | | 44.536 | | 1.110 | | 46.776 | | |
| AUTOPARLLM-CodeGen-16B | **96.760** | 4.42% | **1.173** | 7.50% | **0.882** | 3.63% | **2.260** | 0.89% | 46.040 | -3.27% | **1.100** | 0.91% | **42.640** | 9.69% | **3.4%** |

Apart from the metric scores, we also evaluate the performance of the generated codes by measuring their execution time. We execute an application if at least one loop from that application belongs to our test set of 90 loops. Out of 8 applications, seven applications have at least one loop in the test set. Only EP application has all its loops in the training set. So, we exclude EP from the execution. For all other seven applications, we replace the sequential loops in the testing set with the parallel loops generated using both regular LLMs and AUTOPARLLM augmented LLMs and then

execute the application to measure execution time. We generate the execution time by executing each application five times and then report the average execution time. We use the following formula to calculate the percentage of speedup improvement of AUTOPARLLM augmented LLM generated codes with respect to codes generated by regular LLMs.

$$Speedup\% = (\frac{Execution\_time\_LLM-basic}{Execution\_time\_\text{AUTOPARLLM}\_augmented\_LLM} - 1) * 100$$

We calculate the percentage of speedup for every application and then report the average speedup for all applications in Table 2. Bold fonts show better runtime. It can be observed that codes generated with AUTOPARLLM guidance have 1.9% better runtime than GPT-3.5, 2.6% better runtime than GPT-4, 2.9% better runtime than CodeLlama-34B and 1.4% better runtime than CodeGen-16B in average across all applications in NAS Parallel Benchmark. All experiments are performed on computing nodes with the same configuration. Each computing node has an Intel Xeon Gold 6244 CPU with 32 cores and 366 GB of RAM.

## 5.3 EVALUATING CODE GENERATION ON RODINIA BENCHMARK

We further apply AUTOPARLLM on four applications of Rodinia Benchmark that our GNN models have not seen at all. These applications are developed targeting heterogeneous computing. We extracted 15 loops from these applications using the method described earlier. Out of these 15 loops, 12 loops contain the "private" clause, and three loops contain the "reduction" clause. Then we apply AUTOPARLLM to detect parallelism and pattern of these 15 loops. Out of the 15 loops, AUTOPARLLM is able to correctly detect and classify all the 12 loops with "private" clauses and three loops with "reduction" clauses. As already described in the NAS Benchmark experiments, we create the GNN-guided prompts after generating the predictions. Then, the LLMs are invoked with the GNN-guided prompts to generate the parallel codes based on the patterns. After generating the parallel code, we evaluate the quality of the codes generated by both basic prompt LLMs and AUTOPARLLM in terms of the metrics score as well as our proposed OMPScore. Table 3 shows the results. We can see that AUTOPARLLM guidance results in a better code generation. We also see that AUTOPARLLM has a significantly higher OMPScore for this dataset, too. We also evaluate the runtimes of AUTOPARLLM augmented LLM generated code with regular LLM generated code for this dataset. The runtimes are calculated similarly by executing each application five times and then taking the average on the same computing environments. Table 4 shows the results. The bold fonts show the better runtime. It can be observed that for all applications, codes generated with AUTOPARLLM guidance have better runtimes than the ones generated by basic LLMs. We also calculate the speedup(%) using the same formula introduced earlier. We can see that our AUTOPARLLM augmented LLM generated codes have 1.9%, 2.6%, 2.9%, and 1.4% better runtimes than GPT-3.5, GPT-4, CodeLlama-34B and CodeGen-16B, respectively.

Table 3: Results on Rodinia-3.1 Benchmark Suite (higher indicates better)

| Model | BLEU | CodeBLEU | Rogue-L | METEOR | CodeBERTScore | ParaBLEU | OMPScore |
|---|---|---|---|---|---|---|---|
| CodeGen-16B | 29.87 | 62.34 | 48.57 | 48.19 | 79.0 | 28.52 | 52.88 |
| AUTOPARLLM-CodeGen-16B | **36.93** | **62.77** | **58.67** | **59.97** | **82.2** | **41.73** | **64.57** |
| GPT-3.5 | 56.60 | 81.84 | 75.38 | 71.89 | 91.0 | 46.98 | 79.37 |
| AUTOPARLLM-GPT-3.5 | **71.51** | **89.06** | **91.39** | **85.25** | **97.6** | **68.36** | **98.1** |
| GPT-4 | 55.19 | 81.61 | 76.19 | 70.33 | 91.2 | 49.14 | 79.37 |
| AUTOPARLLM-GPT-4 | **74.83** | **88.03** | **91.39** | **89.58** | **98.6** | **69.09** | **98.1** |
| CodeLlama-34B | 50.08 | 67.82 | 63.50 | 63.07 | 85.9 | 41.92 | 69.55 |
| AUTOPARLLM-CodeLlama-34B | **68.47** | **83.98** | **88.44** | **77.90** | **96.4** | **66.13** | **95.27** |

Table 4: Application-wise execution results (in seconds) on Rodinia Benchmark Suite. Execution times are average of 5 runs.

| Model | BFS | Speedup(%) | B+Tree | Speedup(%) | Heartwall | Speedup(%) | 3D | Speedup(%) | Avg. Speedup(%) |
|---|---|---|---|---|---|---|---|---|---|
| GPT-3.5 | 0.175 | | 3.934 | | 609.809 | | 10.327 | | |
| AUTOPARLLM-GPT3.5 | **0.169** | 3.55% | **3.814** | 3.15% | **603.940** | 0.97% | **10.294** | 0.32% | 1.9% |
| GPT-4 | 0.173 | | 3.931 | | 607.527 | | 10.309 | | |
| AUTOPARLLM-GPT-4 | **0.162** | 6.79% | **3.822** | 2.85% | **603.93** | 0.59% | **10.281** | 0.27% | 2.6% |
| CodeLlama-34B | 0.176 | | 3.936 | | 607.034 | | 10.339 | | |
| AUTOPARLLM-CodeLlama-34B | **0.163** | 7.98% | **3.832** | 2.71% | **603.450** | 0.59% | **10.326** | 0.13% | 2.9% |
| CodeGen-16B | 0.176 | | 3.938 | | 611.284 | | 10.432 | | |
| AUTOPARLLM-CodeGen-16B | **0.174** | 1.15% | **3.836** | 2.65% | **606.492** | 0.79% | **10.349** | 0.80% | 1.4% |

## 5.4 HUMAN EVALUATION RESULTS

For human judgment score we hired two software developers, each boasting over five years of experience in working with OpenMP, meticulously review both the actual and predicted directives generated by each model. For each data point, they allocate scores ranging from 0 (indicative of the lowest quality) to 5 (indicative of the highest quality) to the predicted directive based on the

number of operations needed to transform the predicted directive into the actual directive. Each modification operation results in a deduction of 1 point from the judgment score. Subsequently, the two evaluators engage in discussions to reconcile their assessments and reach a consensus on the final human judgment scores. Secondly, we gauge the correlation between the metrics and human judgment scores using the Spearman ranking (Myers & Sirois, 2004). Specifically, we employ the directives generated using AutoParLLM-GPT-4 with the NAS Parallel Benchmark (NPB) for our human judgement, and the average human score for this set is 4.72. The outcomes of this experiment can be observed in Table 5. Notably, OMPScore achieved the highest correlation with human judgment, reaching up to an impressive 99.99%. This significantly surpasses other existing metrics, exceeding their correlations by up to fourfold. Our evaluation highlights that even the optimized version of the BLEU score like the CodeBLEU (Ren et al., 2020) exhibited the lowest correlation with human judgment. These findings underscore the effectiveness of OpenMPScore over existing metrics when it comes to evaluating OpenMP directives.

Table 5: Comparison of the Spearman Correlation between original metrics and their corresponding metrics provided by OMPScore on the results of NPB benchmark with AutoParLLM-GPT-4

| Correlation | BLEU | CodeBLEU | Rouge-L | Meteor | CodeBERTScore | ParaBLEU | OMPScore |
|---|---|---|---|---|---|---|---|
| Spearman | 24.45 | 28.06 | 61.04 | 24.01 | 48.72 | 22.55 | 99.99 |

## 6 ABLATION STUDY

In this section, we discuss the difference between the parallel versions of the loops generated using basic OMP prompt and using AUTOPARLLM. The generated code of GPT-3.5 and GPT-4 for Listing 1 before and after applying the AUTOPARLLM guidance are shown in Listing 4 and 5 respectively. We can see the latter code generated with AUTOPARLLM is properly decorated with "reduction" clause however without AUTOPARLLM both LLMs fail to add the "reduction" clause.

Listing 4: Listing 1 parallelized by GPT-3.5 and GPT-4

```
#pragma omp parallel for
for (i = 1; i <= 23; i += 1){
    R23 = 0.50 * R23;  T23 = 2.0 * T23; }
```

Listing 5: Listing 1 parallelized by GPT-3.5 and GPT-4 after applying AUTOPARLLM guidance

```
#pragma omp parallel for reduction(*:, R23, T23)
for (i = 1; i <= 23; i += 1){
    R23 = 0.50 * R23;  T23 = 2.0 * T23; }
```

Also, the non-parallel loop in Listing 2 which was wrongly classified as parallel loop by GPT-3.5, GPT-4, CodeGen-16B can correctly classify the loop as non-parallel with the guidance from AUTOPARLLM. The private loop in Listing 3 is also properly parallelized using the "private" clause by all four LLMs after AUTOPARLLM guidance is applied as can be seen in Listing 6 and 7 which also matches with the clause added by developers. So, all three loops in the Motivation section are successfully parallelized using our GNN-enhanced prompt-based AUTOPARLLM approach by mitigating the shortcomings of regular LLMs.

Listing 6: Listing 3 parallelized by GPT-3.5, GPT-4, CodeLlama-34B and CodeGen-16B

```
#pragma omp parallel for
for (i = public.endoPoints;
i <= public.allPoints -1; i += 1) {
    private[i].point_no = i - public.endoPoints;
    //rest of the code }
```

Listing 7: Listing 3 parallelized by all four LLMs after applying AUTOPARLLM guidance

```
#pragma omp parallel for  private (i)
for (i = public.endoPoints;
i <= public.allPoints -1; i += 1) {
    private[i].point_no = i - public.endoPoints;
    //rest of the code }
```

## 7 CONCLUSION AND FUTURE DIRECTION

In this paper, we presented AUTOPARLLM. AUTOPARLLM is an approach built on top of GNNs and LLMs to enable automatic parallelization of programs via inserting OpenMP constructs. The key idea of AUTOPARLLM is that it leverages GNNs to learn control flow and data flow and steers the code generation of LLMs to add appropriate and necessary OpenMP constructs through a specially designed prompt called GNN-guided OMP prompt. Experimental results indicate that the proposed approach is indeed effective, and without GNN guidance, LLM models could produce erroneous or inefficient parallel code. We have plans to extend the approach further to support more types of patterns. Moreover, one interesting extension of our approach is to include some auto-tuning information in our prompt by training the GNN further to predict the number of threads for example, or the loop scheduling types.

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

# 8 APPENDIX

## 8.1 COMPARING WITH TRADITIONAL TOOLS

We further compare the parallelism discovery of AUTOPARLLM against two popular parallelization tools: AutoPar and DiscoPoP. AutoPar performs static analysis to find parallel regions (such as loops), whereas DiscoPoP is a dynamic tool and it executes the code to identify potential parallel regions. Therefore, choosing these two tools enable us to compare AUTOPARLLM against both static and dynamic tools.

We run our GNN-based classifiers to detect the parallel loops in two subsets of the OMP_Serial dataset. These two subsets are specifically developed to compare the performance of parallelism detection models against these two tools, AutoPar and DiscoPoP. The subsets are called AutoPar Subset and DiscoPoP subset. The performance of AutoPar and DiscoPoP in the subsets are quoted from TehraniJamsaz et al. (2023). We run the parallelism detection module of AUTOPARLLM for both of these subsets and report the results in tables 6 and 7 . It can be observed that AUTOPARLLM is quite good at parallelism discovery and achieves 36% higher accuracy than DiscoPoP and 54% higher accuracy than AutoPar. We can also see that both traditional tools have very high precision scores. This is because they are very conservative while detecting parallel loops. Even though, they have high precision scores they miss out on a considerable number of parallelism opportunities.

Table 6: Parallelism Discovery on DiscoPoP Subset (Detecting parallel loops)

| Tool | Precision | Recall | F1-score | Accuracy |
|------|-----------|--------|----------|----------|
| DiscoPoP | 1 | 0.54 | 0.70 | 0.63 |
| AUTOPARLLM | 0.99 | 0.99 | 0.99 | 0.99 |

Table 7: Parallelism Discovery on autoPar Subset (Detecting parallel loops)

| Tool | Precision | Recall | F1-score | Accuracy |
|------|-----------|--------|----------|----------|
| autoPar | 1 | 0.14 | 0.20 | 0.38 |
| AUTOPARLLM | 0.93 | 0.92 | 0.92 | 0.92 |

## 8.2 DETAILS REGARDING THE RGCN MODEL

We used the DGL-based RGCN model[2]. As our RGCN is based on heterogeneous graphs we used 6 HeteroGraphConv laters. In each HeteroGraphConv layer used "sum" aggregation. Relu is used

---

[2]Code for model: https://anonymous.4open.science/r/Project-A-AE4A/README.md

as the activation function. The hidden layer dimension is set to 64. We train our model on batch size = 100. Also, Adam optimizer is used. For loss calculation we used the cross entropy loss function from PyTorch.

## 8.3 GRAPH REPRESENTATION AND FEATURES

Each loop is first converted to IR, and then from that IR, we create the PerfoGraph representation of that loop. There are 3 types of nodes:

- Control node: Each control node represents a statement in IR. Tokens in the IR statement are considered as features for the control node. We embed each token of the statement and finally concatenate the embedding of all tokens to generate the final embedding for a control node.
- Variable node: PerfoGraph contains only the type of a variable in variable nodes. So, we consider the type as the feature for variable nodes and generate embedding for the type-token.
- Constant nodes: For constant nodes, PerfoGraph representation contains both type and value, so we consider both of them as features. First, we generate the embedding for the type-token. For generating the embedding for value-token, we use the Digit Embedding as described in PerfoGraph paper. Finally, the type-token and value-token embeddings are concatenated to generate the final embedding for each constant nodes.

Also, there are three types of edges:

- Control edges: Represents the flow of the program.
- Data edges: Represents the data dependencies of different nodes in the program.
- Call edges: Represents the functional call dependencies of the program.

The nodes are connected with each other using these three different edges. The variable and constant nodes represent the variables and constants that are associated with those IR statements (control nodes). All the embeddings mentioned are generated using the default Pytorch learnable embedding mechanism.

## 8.4 ENSURING CORRECTNESS

When the GNN predicts that a given loop is not parallel AUTOPARLLM does not generate any parallel code for that. So, in that case, AUTOPARLLM outputs the original sequential loop. Hence correctness is preserved. However, like any other ML models there will be False Positives and False Negatives. Let us discuss how we handled those for NAS benchmark, as it contains a bigger test set. During our experiments with NAS benchmark, 30 out of 32 non-parallel loops are correctly detected by AUTOPARLLM. These 30 loops that are detected as non-parallel do not need to be checked further. Because we know that even if there are some parallel loops in this set, treating them as sequential loops will only hurt performance but not correctness. There are 90 loops in the test set of NAS benchmark. That means using AUTOPARLLM, we can filter out 33.33% of loops for analysis safely. Hence developers' workload is reduced by 33.33%. The remaining loops need to be analyzed further after applying AUTOPARLLM, as they may contain the following scenarios:

- Non-parallel loop detected as parallel: Only 2 non-parallel loops are wrongly detected as parallel, we use the original sequential version of these 2 loops to maintain correctness during execution.
- Wrong OMP clause prediction: Only 4 loops have been decorated with the wrong OMP clauses. We also use the sequential version of these 4 loops to maintain correctness. Note that using proper OMP clause will result in speedup but that will give unfair advantage to AUTOPARLLM while calculating performance gain during execution. Hence, we use the sequential version as AUTOPARLLM failed to detect the right clauses.

So, only 6 out of the 60 analyzed loops required manual fixing. The rest 54 loops are already correctly generated by AUTOPARLLM. Finally, we would like to humbly mention that AUTOPARLLM is developed as an intelligent parallelism assistant for developers but not as a replacement.

Based on our results, we believe it is fair to say that AUTOPARLLM generates parallelized versions of loops with very little human effort and greatly increases developers' efficiency.

## 8.5 EXPERIMENTING WITH CHAIN-OF-THOUGHTS (COT) PROMPTING

We performed experiments with Chain-of-Thoughts prompting on the NAS benchmark test set of 90 loops. We applied COT prompting for GPT-3.5, GPT-4 and CodeLlama-34B. For generating the parallel codes, the prompts in Figure 6, 7, and 8 are applied sequentially to create the Chain of Thoughts. Finally, the results are given in Table 8. It can be observed that although COT prompting improved the baseline LLMs, it falls behind the AUTOPARLLM in terms of all code generation metrics and also OMPScore. It worth noting that, even with prompt in Figure 6, the LLMs still had some difficulties to identify cases where a loop should not be parallel.

```
Analyze the given code, and predict whether the code is
parallelizable or not with OpenMP. \n Output can contain only
"yes" or "no". \n Code: {code} \Output:
```

Figure 6: Prompt for Parallelism Discovery

```
Analyse the given code, and predict what kind of OpenMP pattern
may exist in it. \n Output can be None, private, reduction, or
private and reduction together. \n Code: {code} \n Output:
```

Figure 7: Prompt for Pattern Detection

```
Act as a C++ OpenMP Parallelization Tool. \n You will be given
a code. \n Your task is to parallelize the code using OpenMP
and only output the code. \n Add {clause} to the loop. \n Code:
{code} \n Output code:
```

Figure 8: Prompt for Code Generation

Table 8: Results on NAS Parallel Benchmark Suite for COT prompting (higher indicates better)

| Model | BLEU | CodeBLEU | Rogue-L | METEOR | CodeBERTScore | ParaBLEU | OMPScore |
|---|---|---|---|---|---|---|---|
| GPT-3.5 | 27.11 | 52.57 | 42.83 | 35.16 | 72.3 | 27.44 | 41 |
| COT-GPT-3.5 | 30.42 | 55.09 | 49.53 | 36.38 | 74.2 | 30.72 | 50.71 |
| AUTOPARLLM-GPT-3.5 | **38.83** | **65.24** | **91.70** | **55.09** | **96.4** | **48.28** | **95.15** |
| GPT-4 | 26.15 | 54.81 | 43.34 | 36.96 | 73.8 | 27.49 | 46.4 |
| COT-GPT-4 | 29.37 | 56.13 | 56.86 | 41.05 | 76.5 | 29.97 | 58.06 |
| AUTOPARLLM-GPT-4 | **40.12** | **68.24** | **90.40** | **55.68** | **95.2** | **48.48** | **95.15** |
| CodeLlama-34B | 26.44 | 53.48 | 45.15 | 36.40 | 74.0 | 26.48 | 45.44 |
| COT-CodeLlama-34B | 20.41 | 53.75 | 56.66 | 38.86 | 78.6 | 37.46 | 60.52 |
| AUTOPARLLM-CodeLlama-34B | **36.14** | **64.81** | **90.59** | **54.86** | **96.0** | **47.73** | **94.46** |

## 8.6 EVALUATION OF OMPSCORE AFTER REMOVING COMMON TERMS

To mitigate concerns about the prevalence of directives beginning with '#pragma omp parallel for,' we conducted a re-evaluation of the scores of each translation metric and their correlation with human judgment. Specifically, we removed this trivial sequence of tokens from both predicted and expected directives before each evaluation. The results, presented in Table 9, demonstrate that even with the removal of these trivial tokens, OMPScore still achieves very high and also the best correlation with human judgement.

Table 9: Evaluating OMPScore on NAS benchmark without common terms (#pragma omp parallel for) using AUTOPARLLM-GPT-4 setup

| Correlation | BLEU | CodeBLEU | Rogue-L | Meteor | CodeBERTScore | ParaBLEU | OMPScore |
|---|---|---|---|---|---|---|---|
| Spearman | 26.25 | 45.51 | 61.98 | 21.92 | 48.30 | 45.51 | 99.99 |

### 8.7 REPLICATION OF PARABLEU

Our reproduced ParaBLEU score follows the same idea of Wen et al. (2022) but adjusted for OpenMP directives. The formula of ParaBLEU score is shown in Equation 1 as following:

$$ParaBLEU_{OpenMP} = \alpha * BLEU + \beta * BLEU_{OpenMPkeywords} \tag{1}$$

In this formula, $\alpha$ and $\beta$ are the heuristic parameters that we set both of them as $0.5$ for our evaluation. These ratios specify the contribution of original BLEU score and the weighted BLEU score with highlighting the similarities between n-grams containing our selected OpenMP keywords.

