# OpenReview forum: "AUTOPARLLM: GNN-Guided Automatic Code Parallelization using Large Language Models"
_ICLR.cc/2024/Conference — Submitted to ICLR 2024_

### Official Review · Reviewer_6MQq · 2023-10-30

**Soundness:** 3 good
**Presentation:** 3 good
**Contribution:** 2 fair
**Rating:** 5
**Confidence:** 5

**Summary:**

The paper addresses the standard problem of automatically parallelizing the serial version of the code to exploit the compute power of multi-cores in the relatively recent processors. The proposed method, *AUTOPARLLM*, combines GNN based technique to detect the parallelizable loops and then add only a custom defined OpenMP directives to parallelize the loop.

**Strengths:**

- The paper addresses the problem of automatically parallelizing the source code as well as whether a given loop is parallelizable or not.
- The paper is written reasonably and is easy to follow and understand.
- Most of the recent literature in the sub-domain of automatic-parallelization is covered.
- The OMPScore is also an interesting metric.

**Weaknesses:**

- The problem of OpenMP based auto parallelization is a well studied problem ever since the multi-cores became a common place in the modern CPUs. However, the paper tries to propose a limited version of the problem, that is only a loop can be parallelizable or not, if so just add private and/or reduce OpenMP clauses. Doing that with LLMs seem to be a too smaller of a task.
   - Instead, more interesting would be to include all OpenMP pragmas, loop level scheduling, architecture aware compute efficiencies with hardware aware scheduling strategies, etc (for reference on see [1]). Combing all of these non-trivial loop level parallelizations would be a right task for such large capacity LLMs.
- The difference in the execution times of the vanilla LLM generated parallel code versus AUTOPARLLM-CodeGen/GPT generated parallel code is so small <3.0 % (in the best case), it is difficult to state that the difference is significant. There is no mention of the reported execution times to be an average/median of multiple runs of the generated program from each method.
- Especially, given the availability of OpenMP enabled parallel programs in the open source, why not directly fine-tune an LLM to transform the serial code to parallel version.
- There are approaches in LLM literature to transform one language code to another [2], why not try something similar to this directly to address the problem, of course, it requires some minimal refactoring int eh form of pre-train/fine-tuning these open CodeTransformer models.
- The innovation side of the proposed approach are not appearing that great and convincing. It is simple use of multiple methods combined together, first the use of RGCN and then the LLM.
   - In the methods section the details of loss functions at different components of the proposed method are missing.




## References
1. Chennupati G, Azad RM, Ryan C. Synthesis of parallel iterative sorts with multi-core grammatical evolution. InProceedings of the Companion Publication of the 2015 Annual Conference on Genetic and Evolutionary Computation 2015 Jul 11 (pp. 1059-1066).
2. Zügner D, Kirschstein T, Catasta M, Leskovec J, Günnemann S. Language-agnostic representation learning of source code from structure and context. arXiv preprint arXiv:2103.11318. 2021 Mar 21.

**Questions:**

Please follow the weaknesses section for more details on the questions to be addressed.

## Post Rebuttal
- Raising the score still below the acceptance threshold

---

> ### Author Response · Authors · 2023-11-22
> **Reviewer 4**
>
> **Weakness 1:**
>
> **Challenges of parallelization, weakness of SOTA LLMs:** It is true that we worked on a limited version of the parallelization problem, but it is because we believe finding parallelization opportunities and then generating parallel versions of the program is not a trivial task. In the Motivation Examples in our paper we demonstrated that state-of-the-art LLMs even fail to identify parallel loops in seemingly straightforward cases let alone generating parallel codes.
>
> **Challenges of extending support for all OpenMP clauses:** Also, there are not many open source code available for all the OpenMP pragmas for example pragmas like “simd” that offer vectorization and “target” that offer massive data parallelism through GPU-offloading are difficult to find. Hence extending support for all pragmas is challenging due to data scarcity.
>
> **Observations from (Chennupati et al., 2015) paper:** The mentioned approach needs program execution to generate performant parallel codes, which is challenging for HPC codes as most HPC codes are complex and executing them using different configurations has significant overheads in terms of computational resources. However, AUTOPARLLM does not need to execute programs to generate the parallel version of codes. Also, the training time of AUTOPARLLM is very low (around 17 minutes. See Global Response 4.) AUTOPARLLM is also evaluated on a wider range of applications that solve problems in different domains, unlike the one mentioned, which is evaluated on the same type of applications (only sorting).
>
> **Weakness 2:**
>
> **Execution time:** Please see Global Response 1. We clarified the confusion. All reported times are average of 5 runs.
>
> **Improvements:** Please see Global Response 2. AUTOPARLLM approach achieved as high as **14.75%** speedup when the speedup of individual applications are considered.
>
> **Weakness 3:**
>
> **Finetuning:** Finetuning of LLMs come with significant overhead in terms of computational resources. LLMs require large number of samples to learn. Also, we are not only generating parallel codes but we are also considering performance gain like speeup. Unfortunately there are not many use-able open source OpenMP pragma based programs that fits into this criteria. As programs can be represented efficiently using graphs we used Graph based learning approach to learn the control, data and call flow related characteristics of programs. GNNs are much faster and can learn from small number of data points. The training time is around 17 minutes (See details in Global Response 4). Also, some previous studies like Graph2Par, PERFOGRAPH showed better performance using a Graph-based learning approach than transformer-based sequential approaches on parallelism detection.
>
> **Weakness 4:**
>
> **CodeTransformer:** The referred paper (Zügner et al., 2021) uses source code and AST representation of programs. However, for parallelism detection, not only the structure (AST) but also the control, data, and call flows are extremely important. There was no mention of the above flows being incorporated in CodeTransformer. That is why we use a more recent and fine-grained program representation PERFOGRAPH, which works at the IR level and incorporates all those control, data, and call flow-related dependencies in the representation. Also, the transformer-based approaches require a large amount of data to learn. For example, CodeTransformer is trained on 691,974 samples. However, such a large dataset is not available for the problem of parallelism detection and OpenMP-based parallel code generation. Also, as mentioned above the training time for AUTOPARLLM is very low (around 17 minutes. Please see Global Response 4). But it is not mentioned what is the training overhead of CodeTransformer. We also like to mention again that on a smaller dataset, Graph-based learning approaches like Graph2Par, PERFOGRAPH showed better performance than transformer-based sequential approach like PragFormer on parallelism detection.
>
>
> **Weakness 5:**
>
> **Novelty:** Please see Global Response 5.
>
> **Loss functions:** In all cases, the Cross Entropy-based loss function of PyTorch is used.

---

### Official Review · Reviewer_PQiN · 2023-10-31

**Soundness:** 2 fair
**Presentation:** 3 good
**Contribution:** 2 fair
**Rating:** 5
**Confidence:** 3

**Summary:**

This paper introduces a framework for automatically identifying parallelism opportunities and generating appropriate program pragmas for auto-parallelization. The approach leverages a trained Graph Neural Network (GNN) to determine the most suitable parallelism pattern for a given loop nest, and then integrates the GNN's predictions into the prompt to guide a large language model (LLM) in rewriting the program with pragmas. The paper introduces a novel evaluation metric, OMPScore, for assessing auto-parallelization performance by combining regular expressions and program analysis. Experimental results reveal the superiority of this approach over the baseline method, which parallelizes programs directly using LLMs.

**Strengths:**

+ The paper is clearly written with key concepts explained.
+ The proposed OMPScore is carefully designed for evaluating the performance of auto-parallelization via OpenMP directives.
+ The paper tries to tackle the problems of directive prediction and code generation at the same time to perform auto-parallelization. Such a task is significant and extremely challenging.

**Weaknesses:**

+ The paper uses large language models for code generation. As an auto-parallelizing framework, it is unclear whether we really need such LLMs. It appears that LLMs are primarily utilized for inserting pragmas into the outermost loop of a given loop nest, a task that could potentially be accomplished through simpler means such as loop analysis or pattern recognition. The paper could benefit from a more detailed explanation of the advantages of LLMs. For instance, could LLMs be employed to rewrite the code in a way that offers additional parallelization opportunities?
+ The paper mentions that source-to-source compilers miss a lot of parallelism opportunities due to being overly conservative. However, the paper does not compare the optimization results against these compilers or manually-parallelized programs. Consequently, the true effectiveness of this approach remains uncertain.
+ It seems that the OMPScore can only be applied to auto-parallelization with OpenMP directives. How does it compare to other metrics with parallel semantics (e.g., ParaBLEU[1])? Is there any common design philosophy of metrics targeting different code generation approaches?

[1] BabelTower: Learning to Auto-parallelized Program Translation. International Conference on Machine Learning. PMLR, 2022.

**Questions:**

1.	What are the advantages of adopting LLMs for auto-parallelizing?
2.	How does AUTOPARLLM compare to source-to-source compilers or manually-parallelized programs (e.g., OpenMP version in NPB)?
3.	How does it compare to other metrics with parallel semantics (e.g., ParaBLEU)? Is there any common design philosophy of metrics targeting different code generation approaches?

---

> ### Author Response · Authors · 2023-11-22
> **Reviewer 3**
>
> **Weakness 1, Question 1:**
>
> **Challenges of parallelization, advantages of LLMs:** We can parallelize using OpenMP by inserting pragmas in loops. The insertion of pragmas may seem trivial however, before considering the parallelization of loops, let alone inserting pragmas, the control, data, and call flow-related characteristics need to be analyzed. Traditional loop analysis and pattern recognition-based approaches like AutoPar and DiscoPoP can do parallelization to some extent, but they miss a lot of parallelization opportunities due to being overly conservative. The advantage of LLMs is that, given enough background information regarding the intrinsic parallel characteristics of the loops, they can find more parallelism opportunities. That is why we proposed GNNs to learn the intrinsic control, data, and call flow-related features from programs and develop OMP Prompts that help the LLMs to have the necessary background information regarding the underlying parallel characteristics of source programs. We also showed in Appendix 8.1 that AUTOPARLLM finds more parallelism opportunities than those traditional loop-analysis or pattern recognition-based approaches.
>
> **Weakness 2, Question 2:**
>
> **Comparing with source-to-source (S2S) compiler, and traditional tools:** We compared AUTOPARLLM with both DIscoPoP and AutoPar (S2S compiler)  on the DiscoPoP and AutoPar subset of OMP_Serial dataset in Appendix 8.1 of the main paper for the task of Parallelism Discovery. AUTOPARLLM achieved 36% higher accuracy than DiscoPoP and 54% higher accuracy than AutoPar on detecting parallel loops. This clearly indicates the advantage of adopting LLMs for automatic parallelization. However, LLMs struggle to generate proper parallel codes even for simple programs, as shown in Motivation Examples in our paper. The control, data, and flow-related characteristics of loops learned by our GNN module help the LLMs to have the required background information regarding loops to generate the proper parallel version of the loops.
>
> **Weakness 3, Question 3:**
>
> **Comparing with ParaBLEU:** We compared the ParaBLEU score along with other metrics and updated Table 1 and Table 3 in the paper. Our findings remain consistent with ParaBLEU also along with other metrics as it can be observed from Table 1 and Table 3 that AUTOPARLLM improved the ParaBLEU score of LLMs.
>
> Also, we showed a comparison with ParaBLEU using Chain-Of-Thoughts prompting too (Please see Appendix 8.5 Table 8).
>
> The resulting correlation score of ParaBLEU with human evaluation was as low as 22.55 (Please see updated Table 5), whereas OMPScore shows a 99.99 correlation score with human judgment. This outcome underscores the substantial superiority of OMPScore in evaluating OpenMP directives.
>
> We encountered two challenges that hindered our ability to assess the effectiveness of ParaBLEU in evaluating OpenMP directives. Initially, we sought access to the source code of ParaBLEU by reaching out to the authors, but unfortunately, it had not been made publicly available at that time. Consequently, we endeavored to replicate the concept of ParaBLEU scoring for our evaluation. In this replicated version, we assigned additional weights to highlight OpenMP keywords such as 'for,' 'private,' and 'reduction' in addition to the existing weights assigned to CUDA-related keywords. For more details regarding the formula please see Appendix 8.7. The implementation of ParaBLEU is given in the repository:
> https://anonymous.4open.science/r/Project-A-AE4A/parableu_replicated/readme.md
>
>
> **Design Philosophy:** We adopted a design philosophy similar to that of CodeBLEU and CodeBERTScore when creating OMPScore. Both CodeBLEU and CodeBERTScore carefully analyze the characteristics of target programming languages and provide scores that effectively capture these traits. CodeBLEU incorporates information about essential keywords and the syntactic graph of the code, while CodeBERTScore harnesses the capabilities of BERT-based code representations for scoring. OMPScore is designed to adapt to the specific characteristics of pragma directives, including order-sensitive and order-insensitive clauses. This adaptability allows OMPScore to be compatible with various code generation techniques, not limited to AutoParLLM. Furthermore, we have observed similar characteristics in other programming languages, such as Java. For instance, reordering variables in a variable declaration maintains the same code snippet's semantics, while reordering arguments in a method declaration can fundamentally alter the program. Therefore, we plan to enhance OMPScore to accommodate additional programming languages beyond Domain-Specific Languages in future work.

---

### Official Review · Reviewer_eS8a · 2023-10-31

**Soundness:** 3 good
**Presentation:** 3 good
**Contribution:** 3 good
**Rating:** 5
**Confidence:** 3

**Summary:**

The paper proposes a framework that incorporates LLMs to generate a parallel version of sequential programs. The idea is to first use a Graph Neural Network (GNN) to predict the parallel region and the possible OpenMP (OMP) clauses. This prediction can serve as a hint and can be included in the prompt to help the LLM generate better parallel code. The results demonstrate that the proposed method can achieve an average speedup of 2-3% in the benchmark dataset.

**Strengths:**

1.	The idea of enhancing the prompt to generate better code is simple and reasonable.
2.	Using GNN to predict the parallel region is novel and interesting.
3.	A new metric is proposed to provide a better evaluation for parallel code.
4.	The evaluations show the effectiveness of the proposed methods.

**Weaknesses:**

1.	Graph information is not clear: The authors used GNN to determine the parallel regions and predict the OMP clauses from the data-flow, control-flow, and call graphs. However, it remains unclear how the heterogeneous graphs are constructed and how the features of nodes are determined. Also, it would be better if the authors could provide the graph properties, such as the number of nodes, the number of node types, the number of edges, as well as the training details, such as training time and learning curves.

2.	Correctness of the parallel version: The code generation heavily relies on GNN predictions; however, the training/test accuracy of the GNN predictions is not reported. What if the GNN gives a wrong prediction for the parallel regions or the OMP clauses? Furthermore, not all the parallel code can run correctly. The authors only reported the speedup but didn't mention if the program output matches that of the sequential version.

3.	Improvement is not significant: Although the authors claim that their methods can improve the execution time, a 2-3% speedup is not significant. It would be better to conduct multiple runs and report the mean and variance of the execution time to determine if it can indeed bring improvements. Alternatively, the author can demonstrate in which scenarios the framework can achieve more significant improvements.

4.	Heavily rely on the ability of LLM: Table 4 shows similar improvement across different LLM models. However, it also indicates that only stronger LLMs can generate better parallel code, which means the proposed method has a higher probability of generating suboptimal results when using weaker LLMs.

5.	The ability to handle large programs: It appears that the evaluation only considers a single loop at a time. However, real-world programs may have hundreds of parallel regions. There is no evidence that the proposed method can handle multiple parallel regions at once, which may limit its contribution.

6.	Comparison with Chain of thoughts: The idea of the proposed method is to use another model (GNN) to generate better prompts to guide the LLM in generating better results. The root cause is that LLMs cannot generate good results at once without hints. Therefore, it would be reasonable to compare it with the Chain of Thoughts method [1], which aims to complete a task using multiple intermediate reasoning steps. The GNN prediction part can be replaced with another prompt for the LLMs, and the output can be part of the next prompt. This could potentially yield better results compared to using only the basic prompt and could reduce the time required to construct the graphs and train the GNN model.

7.	New metric OMPScore: The OMPScore is an extension of the Rough-L score, which is measured by the longest common subsequence. However, almost all the OpenMP pragmas share the same subsequence, ``#pragma omp parallel for``. Therefore, it doesn't appear to be a suitable metric. Additionally, the authors claim that the OMPScore metric has a high correlation with human evaluation, with a 99.99% Spearman correlation. Considering the potential human bias, it is not entirely convincing to achieve such a high correlation. It is recommended that the authors provide the details or raw data of the evaluation for both the OMPScore and the human evaluation to support this claim.

[1] Wei, Jason, et al. "Chain-of-thought prompting elicits reasoning in large language models." Advances in Neural Information Processing Systems 35 (2022): 24824-24837.

**Questions:**

The idea is simple and clear, but more details should be provided in the methods. Improving LLM generation by prompt engineering is not novel, as far as this reviewer is concerned. Using GNN to predict the parallel region is novel, but it is not quite convincing. It's hard to determine if it can be reproduced, especially regarding the GNN training, execution time improvement, and human evaluation for the metrics.  Can the authors comment on the novelty of the proposed approach, i.e. how does the proposed approach improve over the state-of-the-art?

---

> ### Author Response · Authors · 2023-11-22
> **Reviewer 2**
>
> **Weakness 1:**
>
> **Heterogeneous Graph Construction, node-types, edge-types, and features:**  Please see Appendix 8.3.
>
> **Training Details, learning curves:** Please see Global Response 4.
>
> **Num of Nodes, edges:** The average number of nodes in the PerfoGraph representation of OMP_serial dataset is 67.39, and the average number of edges in the dataset is 116.05.
>
> **Weakness 2:**
>
> **Correctness:** We ensured that all parallel codes that are generated by LLMs are correct before program execution. Please see Appendix 8.4 where we describe how we handle cases where LLMs wrongly parallelize a loop or generate wrong OMP clauses.
>
> **GNN accuracy:** We reported the GNN prediction accuracy of Parallelism Discovery for NAS benchmark. Also, we reported the number of correctly classified loops of each type for NAS benchmark. Please see Section 5.2 (Page 7). For Rodinia benchmark, we also mentioned the number of correctly classified loops of each type. Please see Section 5.3 (Page 8). However, we also include the summary below.
>
> |              Task                             | NAS Benchmark | Rodinia Benchmark|
> |------------------------------------------|-------------------------|---------------------------|
> |Parallelism Discovery Accuracy |          94.44%       |          100%            |
> |Private Detection Accuracy        |          92.86%       |          100%            |
> |Reduction Detection Accuracy   |          100%          |          100%            |
>
> **Weakness 3:**
>
> **Execution times:** We reported the average execution time of 5 runs for each application in the paper. Please see Global Response 1.
>
> **Improvements:** When speedup of individual applications are considered then AUTOPARLLM achieves as high as **14.75%** speedup.  Please see Global Response 2 for details.
>
> **Weakness 4:**
>
> **Dependence on LLMs:** AUTOPARLLM achieved good speedup with weaker LLMs too. For example, in Table 2 of the paper, it can be seen that AUTOPARLLM with CodeGen-16B achieved the highest 3.4% average speedup for NAS benchmark which is similar to the speedup of the most powerful AUTOPARLLM-GPT-4 model.
>
> **Weakness 5:**
>
> **Handling Multiloops:** Please see Global Response 3. AUTOPARLLM is designed to handle multiple loops.
>
>
> **Weakness 6:**
>
> **COT Prompting:** Please see Table 8 in Appendix 8.5. Due to time constraints, we performed Chain-of-Thought (COT) experiments with three of the LLMs: GPT-3.5, GPT-4, and CodeLlama on the NAS benchmark test set of 90 loops. However, AUTOPARLLM outperformed the COT approach in terms of all code generation metrics. Detailed prompts and results are given in Appendix 8.5.
>
> **Weakness 7:**
>
> **Incorporating ROGUE-L in OMPScore:** Before incorporating ROUGE-L as a component of OMPScore, we conducted an analysis to assess the correlation between several established translation evaluation metrics and human evaluations. The results, presented in Table 5 of our submission, indicated that ROUGE-L exhibited the highest correlation with human scores, as determined by Spearman ranking. We attribute the superior performance of ROUGE-L to a trade-off between the complexities of NLP-based metrics and programming language-based metrics. While NLP-based metrics like BLEU and Meteor primarily emphasize similarity at the word or n-gram levels without considering the alignment between the expected and translated results, metrics like CodeBLEU and CodeBERTScore are designed for object-oriented and scripting languages, which are considerably more intricate compared to domain-specific languages like OpenMP.
>
> **Calculating correlation removing common terms:** Please see Appendix 8.6 for details. We calculated the correlation of OMPScore with human judgment by removing the common terms “#pragma omp parallel for”. OMPScore still achieved a very high and the best Spearman score (99.99) among all other metrics.
>
> **Raw data of Human Evaluation:** All data regarding the Human Evaluation scores are given in the repo.
> https://anonymous.4open.science/r/Project-A-AE4A/Human-Evaluation/AutoParLLM_HumanScore.xlsx
>
> **Questions:**
>
> **Reproducibility:** We have already provided with the codes necessary for training the GNNs (base.py) and doing predictions using the GNNs (main.py) in the anonymous repository (https://anonymous.4open.science/r/Project-A-AE4A/base.py), and we mentioned the repository in the Appendix (Page 12). Also, the codes generated by AUTOPARLLM using all four LLMs are also provided in the repository. The virtual environment dependencies are also given in the requirements.txt file. We also included the file-name, loop-number etc. so that one can easily find from which benchmark and from which application a certain loop is extracted. Both of the benchmarks are open-source, and they have very clear instructions regarding the execution of the applications.
>
> **Novelty:** Please see Global Response 5.

---

> > ### Comment · Reviewer_eS8a · 2023-11-22
> >
> > Thank you very much for the detailed answers to my concerns.

---

### Official Review · Reviewer_pN6K · 2023-11-03

**Soundness:** 3 good
**Presentation:** 3 good
**Contribution:** 4 excellent
**Rating:** 8
**Confidence:** 5

**Summary:**

In AutoParLLM the authors introduce a new guidance technique for prompting large language models for code to parallelize a presented piece of software by inserting OpenMP directives. To better measure the parallelization performance the authors additionally introduce a new evaluation metric called OMP-Score to better capture the preferences of parallelized code. The guiding graph neural network first learns the control flow, data flow, and call flow of the program, before feeding this additional context to the large language model, which is being prompted to parallelize the presented piece of code.

The guidance with graph neural networks produces consistently better results across established evaluation metrics, as well as the new OMP-Score across a set of 4 of the most commonly used large language models for code with CodeGen, CodeLLama, and the commonly used GPT-3.5, and GPT-4. Human evaluation of the OMP-Score with 2 experienced software engineers shows good alignment between the rating through OMP-Score, and the human rating of produced results.

**Strengths:**

With a strong clarity of exposition the strength of the paper lies in its clear story-line, which is complemented by strong evaluations on established benchmarks. Including a strong outperformance of the approach on the BLEU, CodeBLEU, Rogue-L, METEOR, CodeBERTScore, and OMPScore, as well as the more close to practice measurement of execution time.

The clear explanation of intuitions behind decisions, and the GNN-guidance make the paper very easy to understand, and is refreshing in its clarity. In addition the key contribution of the paper, the GNN-guidance to provide context to the LLM-query, is clearly pointed out and illustrated what this means for the prompts in practice.

The strong performance across all benchmarks only backs this contribution up further.

**Weaknesses:**

While the aforementioned analysis of the GNN-guidance for LLMs is very thorough, extensive, and backed up with strong results across the evaluation metrics, and execution time, I believe that the same cannot be extended to the proposed OMPScore metric. As such I would urge the authors to extend the evaluation, and validation of OMPScore as a new evaluation metric either in the core paper, or in the appendix.

In addition, I would urge the authors to include further literature references in related work. As the presented approach relies on the compilation through IR two works that stand out, and should see inclusion are:
- Transcoder-IR --> Code translation with Compiler Representations
- Automap/PartIR --> Automap: Towards Ergonomic Automated Parallelism for ML Models

**Questions:**

- With the strong outperformance of AutoParLLM on the established evaluation metrics, as well as OMPScore, how would the authors explain that the average speed on the execution time is closer to 2-3%?
- Taking CodeLlama as an example, would it be possible to additionally repeat the evaluation for the different sizes of CodeLlama available to verify that the authors claim holds across differently sized models?
- Only having focussed on OpenMP CPU-parallelization, do you see this approach extending to OpenMP device offloading, where there is much less code available?

---

> ### Author Response · Authors · 2023-11-22
> **Reviewer 1 response**
>
> **Weakness:**
>
> **Reference of Transcoder-IR and automap/part:** Thanks for the suggestions. Yes, both of these works are IR based where one uses IR representation for code translation and the other uses IR based representation for parallelizing ML models. We referred the Automap/PartIR paper in Section 2 (Related Works: Data-driven approaches) and the Transcoder-IR paper in Section 4 (Approach: Program Representation) in the paper.
>
> **Question 1:**
>
> **Speedup:** The 2-3% speedup reported in Table 2 and Table 4 represents the average speedup across the applications in the NAS and Rodinia benchmark. For example, from Table 2, we can say that using AUTOPARLLM-GPT-4, we have an average 3.4% speedup than regular GPT-4 based parallelization across seven applications in NAS benchmark. However, if the speedup of individual applications is considered, then AUTOPARLLM achieved as high as 14.75% (for SP application in NAS benchmark) and 7.98% (for BFS application in Rodinia benchmark) speedup. Please refer to the Global Response 2 for the details regarding individual application-based speedups. Also, we updated Table 2 and Table 4 in the original paper.
>
> **Question 2:**
>
> **CodeLlama:** Thanks for this suggestion. Our experiments already show that AUTOPARLLM approach can improve the performance of different-sized models like GPT-3.5 (175B parameters), GPT-4 (1.5T parameters), CodeGen (16B parameters), and CodeLlama (34B parameters). However, we believe that it is definitely possible to experiment with the different sizes of CodeLlama. We will include the experiments with different-sized CodeLlama in our final camera-ready submission.
>
> **Question 3:**
>
> **Extend to GPU-offloading:** Yes, actually, in future works, we mentioned that our goal is to include more patterns for parallelization. For example, parallelizing Stencil patterns. Also, we have plans to include support for pragmas that support GPU-offloading-based parallelism (e.g., “target” clause in OpenMP). However, as rightly mentioned finding open-source code for these patterns is not trivial. We have plans to extract templates from these patterns and then create synthetic samples for training our data-driven models.

---

> > ### Comment · Reviewer_pN6K · 2023-11-23
> > **Thank you for addressing my comments**
> >
> > I thank the authors for their effort in addressing my comments. I feel that the score of 8 is validated.

---

### Author Response · Authors · 2023-11-22
**Global Response**

We would like to thank the reviewers for their comments. Here we tried to address some of the common concerns raised by the reviewers.

**(1) Execution times are average of 5 runs:** Actually, we mentioned that the applications are executed five times and average of the execution times are reported for both NAS and Rodinia Benchmark.

Please see Section 5.2 Page 8. Quoted from paper:
“ We generate the execution time by executing each application five times and then report the average execution time. ”

Also, see Section 5.3 Page 8. Quoted from paper:
“ The runtimes are calculated similarly by executing each application five times and then taking the average on the same computing environments. ”

However, we believe it was not noticeable easily, so we also give this information in Table 2 and Table 4 caption in the updated submission.

**(2) Speedup Improvements:** The speedup reported for NAS benchmark is the average speedup across the 7 applications in NAS and speedup reported for Rodinia benchmark is the average speedup across the 4 applications in Rodinia. However, we believe that it is more appropriate to show the speedup of individual applications because each application is different, they serve different purposes, and the applications are also executed separately.  We updated Table 2 with Table 4 of original paper to show speedup of individual applications in NAS and Rodinia Benchmark. Now It can be observed that AUTOPARLLM approach achieved as high as an impressive **14.75%** speedup for SP application of NAS benchmark and as high as **7.98%** speedup in the BFS application of Rodinia benchmark.
Based on the above results, we believe it is fair to say that AUTOPARLLM achieved significant speedups in multiple applications of the two benchmarks.

**(3) Handling Multiloops:** Most of the applications that we tested are quite large and have multiple loops. Number of loops for each application in our test set is given below for NPB and Rodinia. AUTOPARLLM can handle multiple loops. Only the user needs to provide a code that is compileable because it uses the IR representation of the code. AUTOPARLLM will run the Rose outliner for extracting loops, then it will use “llvm-extract” to extract the loop-specific IR, then it will generate predictions for the loops and invoke LLMs to finally generate the parallel version of the loop.

**NPB**
| Application | # of loops |
|-------------|------------|
| BT          | 7          |
| IS          | 2          |
| CG          | 2          |
| FT          | 5          |
| LU          | 17         |
| MG          | 17         |
| SP          | 40         |

**Rodinia**
| Application | # of loops |
|-------------|------------|
| BFS         | 1          |
| B+Tree      | 4          |
| Heartwall   | 9          |
| 3D          | 1          |

**(4) Training details, learning curves, code, dataset:** The training time for each GNN model is given below. For reporting training time, each model is trained five times and the average training time is reported.

*Parallelism Detection model: 17 min 5 sec*

*Private Detection model: 9 min 27 sec*

*Reduction Detection model: 9 min 12 sec*

The dataset and models are provided in the anonymous repo

(https://anonymous.4open.science/r/Project-A-AE4A/main.py ).

The learning curves for all three models are given in the repo.

(https://anonymous.4open.science/r/Project-A-AE4A/training_graphs/red-non-red.png).

**(5) Novelty:** The Motivation section (Section 3) shows the weakness of LLMs in identifying and generating parallel code due to not having proper background information regarding programs. We improved the ability of LLMs to generate better parallel codes. Below we list the main novelties of our work:

- We believe this is the first work that incorporates the control, data, and call flow-related characteristics of programs in the LLMs’ prompts by using guidance from GNNs.
- AUTOPARLLM improved the capacity of LLMs to generate parallel codes both in terms of well-known code generation metrics and speedup.
- AUTOPARLLM also outperforms the Chain-of-Thoghts (COT) prompting technique. (Please see Appendix 8.5)
- Also, we proposed OMPScore, a new metric for evaluating OMP-based parallel codes, which shows a better correlation with human judgment than other established metrics.

---

### Meta-Review · Area_Chair_tynr · 2023-12-06

**Metareview:**

The paper proposes doing automatic parallelization using a deep learning pipeline where a GNN first analyses the input source code, followed by a LLM that emits OMP pragmas. They find a modest speedup in the generated code over directly prompting LLMs to produce pragmas, and find that they have higher recall, but lower precision, compared to classic static and dynamic analyses (so a human user needs to inspect the pragma to make sure that nothing is being incorrectly parallelized).

A strength of the work is that it is empirically rigorous, considering many metrics and 2 benchmarks. The primary downside is it offers only modest runtime improvement over simply prompting an LLM, and obtaining that improvement seems important, given it does not have the correctness guarantees of classic methods.

It is possible that this paper would be more successful in a conference that caters to parallelization or program analysis, as the primary result appears to be a bespoke machine learning model for applications in those areas.

**Justification For Why Not Higher Score:**

Given the complexity of the method, it really should have better empirical results vs just LLM prompting. Given the bespoke application, it should cover a broader range of parallelization strategies (varieties of omp pragmas).

**Justification For Why Not Lower Score:**

n/a

---

### Decision · Program_Chairs · 2024-01-16

Reject